# Caspase-8 contributes to angiogenesis and chemotherapy resistance in glioblastoma

Giulia Fianco[1,2], Maria Patrizia Mongiardi[3], Andrea Levi[3], Teresa De Luca[4], Marianna Desideri[4], Daniela Trisciuoglio[4], Donatella Del Bufalo[4], Irene Cinà[2], Anna Di Benedetto[5], Marcella Mottolese[5], Antonietta Gentile[6,7], Diego Centonze[6,7], Fabrizio Ferrè[8], Daniela Barilà[1,2]*

[1]Department of Biology, University of Rome Tor Vergata, Rome, Italy; [2]Laboratory of Cell Signaling, Istituto di Ricovero e Cura a Carattere Scientifico (IRCCS) Fondazione Santa Lucia, Rome, Italy; [3]Institute of Cell Biology and Neurobiology, Consiglio Nazionale delle Ricerche (CNR), Rome, Italy; [4]Preclinical Models and New Therapeutic Agents Unit, Research, Advanced Diagnostics and Technological Innovation Department, Regina Elena National Cancer Institute, Rome, Italy; [5]Pathology Department, Regina Elena National Cancer Institute, Rome, Italy; [6]Multiple Sclerosis Clinical and Research Center, Department of Systems Medicine, University of Rome Tor Vergata, Rome, Italy; [7]Unit of Neurology and of Neurorehabilitation, IRCCS Istituto Neurologico Mediterraneo (INM) Neuromed, Pozzilli (IS), Italy; [8]Department of Pharmacy and Biotechnology (FaBiT), University of Bologna, Bologna, Italy

*For correspondence: daniela.barila@uniroma2.it

Competing interests: The authors declare that no competing interests exist.

**Abstract** Caspase-8 is a key player in extrinsic apoptosis and its activity is often downregulated in cancer. However, human Caspase-8 expression is retained in some tumors, including glioblastoma (GBM), suggesting that it may support cancer growth in these contexts. GBM, the most aggressive of the gliomas, is characterized by extensive angiogenesis and by an inflammatory microenvironment that support its development and resistance to therapies. We have recently shown that Caspase-8 sustains neoplastic transformation in vitro in human GBM cell lines. Here, we demonstrate that Caspase-8, through activation of NF-kB, enhances the expression and secretion of VEGF, IL-6, IL-8, IL-1beta and MCP-1, leading to neovascularization and increased resistance to Temozolomide. Importantly, the bioinformatics analysis of microarray gene expression data derived from a set of high-grade human gliomas, shows that high Caspase-8 expression levels correlate with a worse prognosis.

## Introduction

The downregulation of apoptotic pathways is a hallmark of cancer (*Hanahan and Weinberg, 2011*). Caspase-8 is a central player in the apoptotic cascade triggered by death receptors stimulation (*Juo et al., 1998*); consistently, its expression (*Pingoud-Meier et al., 2003*) or its apoptotic activity (*Cursi et al., 2006*; *Safa et al., 2008*) are often reduced in cancer. The observation that Caspase-8 is retained in many tumors (reviewed in [*Stupack, 2013*]) suggests a dual role for Caspase-8 in cancer. The identification of several non-canonical functions of Caspase-8 that are independent of its enzymatic activity and of apoptosis, supports this idea. Indeed, Caspase-8 modulates cell

**eLife digest** Cancer cells are different to normal cells in various ways. Most cancer cells, for example, delete or switch off the gene for a protein called Caspase-8. This is because this protein is best known for promoting cell death and stopping tumor cells from growing. However, some cancers keep the gene for Caspase-8 switched on including glioblastoma, the most aggressive type of brain cancer in adults. This begged the question whether this protein may in fact promote the development of tumors under certain circumstances.

Glioblastomas are often highly resistant to chemotherapy and can communicate with nearby cells using proteins called cytokines to promote the formation of new blood vessels. The new blood vessel allows the tumor to readily spread into healthy brain tissue, which in turn makes it difficult for surgeons to remove all the cancerous cells. As a result, glioblastomas almost always return after surgery, and so there is strong need for new effective treatments for this type of cancer.

Fianco et al. have now investigated whether Caspase-8 helps glioblastomas to grow and form new blood vessels. One common method to study human cancer cells is to inject them into mice and watch how they grow, because these experiments mimic how tumors develop in the human body. When mice were injected with human glioblastoma cells with experimentally reduced levels of Caspase-8, the cells grew poorly and did not form as many new blood vessels as unaltered glioblastoma cells. Further experiments showed that, when grown in the laboratory, glioblastoma cells with less Caspase-8 were more sensitive to a chemotherapeutic drug called temozolomide. These findings confirm that Caspase-8 does boost the growth and drug resistance of at least one cancer. When Fianco et al. analyzed clinical data from patients affected by glioblastoma, they also observed that those patients with high levels of Caspase-8 often had the worse outcomes.

Previous studies conducted in white blood cells showed that Caspase-8 activated a protein complex called NF-kB, which in turn led to the cells releasing cytokines. Fianco et al. have now verified that Caspase-8 promotes NF-kB activity also in glioblastoma cells, and that this causes the cancer cells to release more cytokines. As such, these findings reveal a clear link between Caspase-8 and the formation of new blood vessels by glioblastomas.

Future studies are now needed to understand why Caspase-8 promotes cell death in some cancers but the formation of new blood vessels in others. Indeed, Caspase-8 might become a target for new anticancer drugs if it is possible to inhibit its cancer-boosting activity without interfering with its ability to promote cell death.

adhesion and migration, suggesting that in cancer cells Caspase-8 may be rewired from apoptosis to alternative pathways that sustain tumor growth (reviewed in [*Graf et al., 2014*]).

We recently demonstrated that Caspase-8 promotes the proliferation and neoplastic transformation of glioblastoma (GBM) cell lines (*Fianco et al., 2016*). Interestingly, large-scale gene expression approaches have demonstrated *Caspase-8* upregulation in GBM compared to normal tissue; in particular, the mesenchymal subtype of GBMs is characterized by high *Caspase-8* expression (*Verhaak et al., 2010*).

The fatal nature of GBM is strongly associated with its extensive angiogenesis (*Kargiotis et al., 2006*), and with its capacity to infiltrate throughout the brain tissue and to resist to chemotherapy (*Dunn et al., 2012*).

Tumor neoangiogenesis is strongly supported by an inflammatory microenvironment that also promotes the proliferation of tumor cells and the survival of malignant cells and alters responses to chemotherapeutic agents (*Mantovani et al., 2008*). Consistently, in vitro and in vivo studies have identified high levels of IL-8, IL-6 and IL-1beta in the conditioned media (CM) of several GBM cell lines and in microenvironment of clinical samples (reviewed in *Yeung et al., [2013]*). This often depends on overactive EGFR signalling, which stimulates NF-kB, AP-1 and cEBP transcription factors, thereby promoting the expression of IL-8 and IL-6 (*Bonavia et al., 2012*; *Inda et al., 2010*).

The work of several laboratories has identified Caspase-8 as an activator of NF-kB in B cells downstream of antigen receptors (*Su et al., 2005*) and Toll-like receptors (*Lemmers et al., 2007*), as well as in T cells (*Bidère et al., 2006*). These observations, along with the pivotal role of NF-kB in

modulating cytokine production, in shaping tumor microenvironment and in promoting angiogenesis and GBM progression (reviewed in *Karin et al. [2002]*, *Dunn et al. [2012]*, *Yeung et al. [2013]* and *Nogueira et al. [2011]*), prompted us to investigate whether high *Caspase-8* expression in GBM promotes these functions.

## Results and discussion

To investigate the possible role of Caspase-8 in GBM angiogenesis, we sub-cutaneously injected mice with matrigel-containing conditioned media (CM) from U87MG (U87) cells, in which Caspase-8 expression was genetically silenced (shC8) or not (shcontrol, named CTR) (as shown in *Figure 1—figure supplement 1*). Matrigel plugs containing CM from U87CTR induced a strong angiogenic response as evidenced by macroscopic analysis and haemoglobin content, similar to that detected in the positive control where VEGF has been added to the media. Importantly, matrigel plugs containing CM from U87 shC8 displayed a significant reduction of both angiogenesis in vivo and the haemoglobin content (*Figure 1A and B*).

In agreement with this finding, CM from U87shC8 cells was 2.5-fold less potent in triggering the proliferation of Human Umbilical Vein Endothelial Cells (HUVECs) compared to CM from U87CTR cells (*Figure 1—figure supplement 2*). Moreover, when exposed to CM from U87CTR cells, HUVECs formed tube-like structures resembling a capillary plexus; conversely, partially organized and rounded endothelial cells were observed after the addition of CM from U87shC8 cells (*Figure 1—figure supplement 2*).

To uncover whether Caspase-8 dependent regulation of angiogenesis correlates with the modulation of tumor growth in vivo, we compared the tumorigenic potential of U87 cells in which the expression of Caspase-8 was genetically inhibited or not. As reported in *Figure 1C*, after three weeks, U87 cells formed tumors of an average volume of ~1,254 $mm^3$ in all injected mice, whereas U87shC8 cells exhibited a drastically reduced capacity to form tumors (mean tumor volume: ~148 $mm^3$), which were detected in only 29% of injected mice. After 6 weeks, the mean tumor volume generated by U87sh8 cells was ~344 $mm^3$ and tumors were identified in 67% of injected mice (*Figure 1C*).

In order to evaluate whether the decreased tumor growth observed in shC8 mice compared to CTR was associated with a lower vascular density, we evaluated neovascularization in CTR and shC8 tumors. Tumor vessels were visualized using anti-CD31 mAb, which permitted an assessment of the vascular density of the tumors. *Figure 1D* shows that there was a significant difference in vascularization between CTR and shC8 tumors; the mean number of vessels per $mm^2$ field was 20.52 (±0.62) for CTR and 1.21 (±0.41) for shC8 (p=0.001).

Overall, these data suggest that Caspase-8 expression in GBM is required for tumor growth, maximal proliferation of GBM-associated endothelial cells and angiogenesis, probably through the production of secreted factors.

To measure whether Caspase-8 expression affects the profile of secreted cytokines and growth factors, we used LUMINEX multiplex bead assay, which allows the simultaneous quantification of several cytokines from the same sample (*Khalifian et al., 2015*). High levels of IL-1$\beta$, IL-6, IL-8, MCP-1, VEGF-A and TNFalpha were present in the CM of U87CTR, as is consistent with previous reports (*Albulescu et al., 2013*), and the genetic inhibition of Caspase-8 caused a strong decrease in the levels of these cytokines (except TNF alpha) in the CM (*Figure 2A*). Accordingly, we observed a dramatic reduction of the respective mRNAs upon Caspase-8 downregulation (*Figure 2B*). We obtained similar results in U87 cells where Caspase-8 expression was inhibited using a different interfering sequence (*Figure 2—figure supplement 1*), as well as in U251, another GBM cell line (*Figure 2—figure supplement 1*).

To clarify the significance of our findings, we interrogated mRNA levels in a RNA-seq data set derived from 174 human GBM patients, which is available in Cancer Genome Atlas (TCGA). For each gene, we built a profile of normalized expression data from the 174 samples, and we then compared these profiles using the Pearson product-moment correlation coefficient. Interestingly, the *Caspase-8* expression profile positively correlated with those of *IL-6, IL-8, IL1$\beta$* and *MCP-1* (*Figure 2C* and *Figure 2—source data 4*), ranging from 0.25 when comparing the *Caspase-8* and *IL-6* GBM expression profiles, to 0.53 for *Caspase-8* and *MCP-1*. All of these correlations were statistically significant

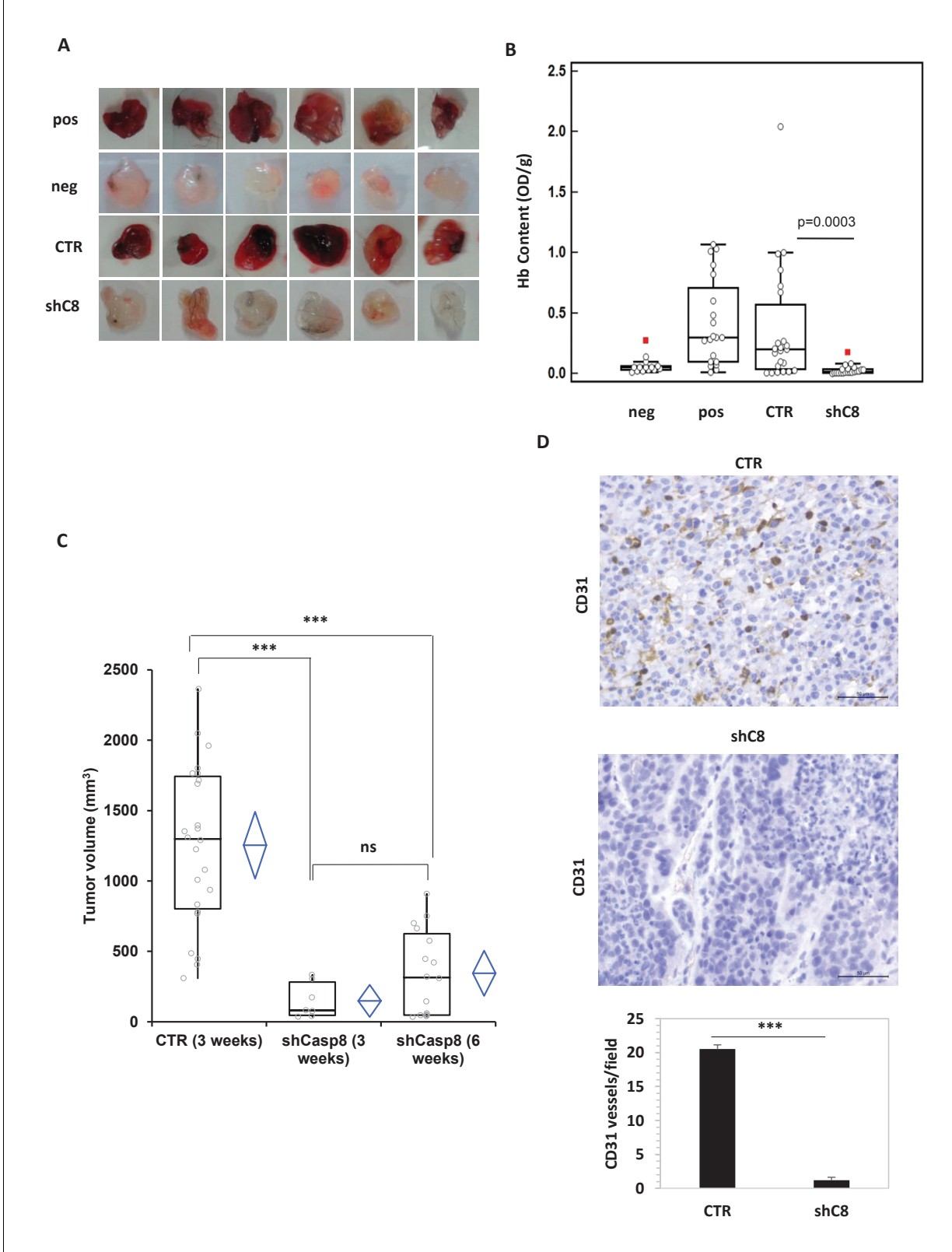

**Figure 1.** Caspase-8 expression promotes tumor growth and neoangiogenesis in vitro and in vivo. (A, B) Caspase-8 expression promotes the ability of conditioned medium (CM) from U87 cells to induce neo-angiogenesis in vivo. Representative images illustrate the macroscopic analysis (A) and quantification of Hb content (B) of Matrigel plugs containing CM from Sh Control (CTR, n = 23) or Sh Caspase-8 (ShC8, n = 21) cells. The negative (Neg, n = 14) and positive (Pos, n = 20) controls contained heparin alone or heparin plus VEGF, respectively. The values of biological replicates (n) for each

*Figure 1 continued on next page*

*Figure 1 continued*

condition are shown as single dot, and are expressed as optical density (OD 540 nm)/g of the Matrigel plug. The Mann-Whitney test (independent samples) was used for statistical analyses. In all experiments, the volume of CM from different samples was normalized on the number of cells for each sample counted when CM was collected. (C) Comparison of the tumor size between U87 CTR and U87shC8 mouse xenografts. Quantitative analysis, by Kruskal-Wallis test with Bonferroni correction, of the volume of tumors measured at 3 and 6 weeks after cell injection. Each plot graphically shows the central location and scatter/dispersion of the values of each group: the line series in the rectangular-shaped boxes indicate the median value of the data and the end of the vertical lines indicate the minimum and the maximum data value. The means and their confidence intervals are shown in the diamond-shaped box. P-value was calculated according to the independent samples t-test. Each dot corresponds to the tumor value of one mouse. ***p<0.001. (D) The microvessel density, determined immunohistochemically by the means of an anti-CD31 antibody recognizing murine endothelial cells, evidenced the presence of a significantly higher number of vessels in CTR cells (evaluated as mean ± SD in CTR tumors) than in shC8 tumors (***p<0.001). Original magnification 40X, scale bar 50 μm.

The following source data and figure supplements are available for figure 1:

**Source data 1.** Caspase-8 mRNA is efficiently silenced in shC8 and shC8#2 cell lines compared to CTR cells.

**Source data 2.** Caspase-8 expression promotes the ability of conditioned medium (CM) from U87 cells to induce neo-angiogenesis

**Source data 3.** Caspase-8 expression promotes tumor growth in mouse xenograft experiments.

**Source data 4.** Caspase-8 expression promotes neovascularization in vivo.

**Figure supplement 1.** Caspase-8 mRNA and protein expression is efficiently silenced in shC8 and shC8#2 cell lines compared to CTR cells.

**Figure supplement 2.** Caspase-8 expression promotes the ability of conditioned medium from U87 cells to induce endothelial cells proliferation and Capillary Tube-Like Network Formation.

(P-value < 0.0008 for each comparison), whereas there was no significant relationship between the *Caspase-8* and *VEGF* expression profiles (Pearson correlation coefficient 0.09, p-value 0.22).

Overall these results identified a novel role of *Caspase-8* in promoting neoangiogenesis. This is in agreement with previous studies on *Caspase-8* null mice, which showed defects in the angiogenesis programme during development (*Scharner et al., 2009*).

The expression of the aforementioned cytokines is finely tuned by several transcription factors, among which is NF-kB. In gliomas, NF-kB is often constitutively activated (*Cahill et al., 2016*), NF-kB-regulated genes are induced (*Bonavia et al., 2012*), and the expression of these genes correlates inversely with patient prognosis (*Nogueira et al., 2011*). Consistently, the overexpression of a dominant negative mutant of IKBalpha (IKBaphaS32A/S36A, named IKBapha SR) that inhibits NF-kB signalling severely decreased the expression of *IL-6, IL-8, IL1β, MCP-1* and *VEGF-A* mRNAs in U87 cells (*Figure 2—figure supplement 2*). Silencing of *Caspase-8* in U87 cells did not reduce the amount of NF-kB but strongly decreased its nuclear localization (*Figure 3A and B*). Similar results were also obtained in U251 GBM cells (*Figure 3C*). Preliminary immunohistochemistry analysis in formalin-fixed paraffin-embedded tissues suggests a different distribution of NF-kB in CTR (nuclear and cytoplasmic) and in shC8 (mainly cytoplasmic) tumors (*Figure 3—figure supplement 1*). Consistently, we could detect a significant reduction of *VEGF* and *IL-8* mRNA levels in shC8-derived tumor samples (*Figure 3—figure supplement 1*). Our results provide the first evidence that Caspase-8 promotes NF-kB activity in GBM in vitro and in vivo. Interestingly, cancer-associated missense mutations of *Caspase-8* resulting in stronger activation of NF-kB have been identified recently in head and neck squamous cell carcinoma (*Ando et al., 2013*).

Having demonstrated that *Caspase-8* expression promotes neoplastic transformation in vitro (*Fianco et al., 2016*) and sustains tumor growth and its microenvironment (*Figures 1*, *2* and *3*), we asked whether *Caspase-8* expression affects GBM cells response to therapy. For several decades, the typical treatment for GBM has been radiotherapy; more recently, Temozolomide (TMZ) has been incorporated into the standard treatment as an essential component (*Stupp et al., 2005*).

As shown in *Figure 4A* and in *Figure 4—figure supplement 1*, the down-regulation of Caspase-8 expression significantly sensitized U87 cells to the cytotoxic effects of TMZ. Interestingly, the expression of the dominant negative mutant of IKBalpha, IKBaphaSR, had a similar effect

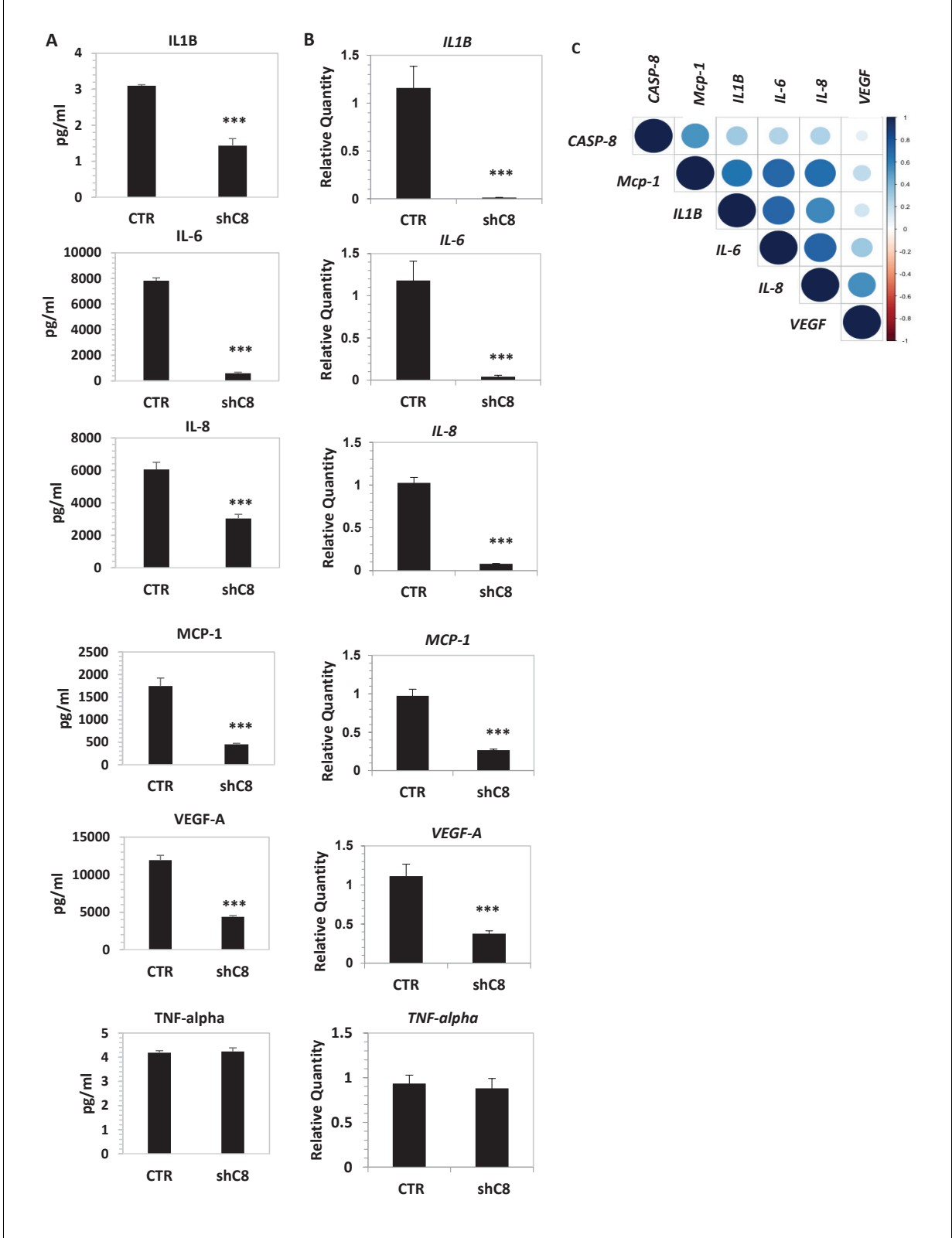

**Figure 2.** Caspase-8 promotes the expression and secretion of cytokines and growth factors. (**A**) Concentrations (pg/ml) of IL-1β, IL-6, IL-8, MCP-1 (CCL-2), VEGF-A and TNF-α were measured by Luminex assay, in the supernatants of U87 cells upon stable genetic silencing of Caspase-8 expression (Sh C8) or not (ShCTR, named CTR). Data were plotted as mean ± SD and statistical significance was estimated by Unpaired T-test, ***p<0.001. Error bars represent a SD between three independent experiments, each of them performed in technical duplicate. In all experiments, the volume of CM

*Figure 2 continued on next page*

*Figure 2 continued*

from different samples was normalized on the number of cells for each sample counted when CM was collected. (**B**) Quantitative real time RT-PCR on U87CTR cells and U87shC8 cells. Relative quantities were calculated normalizing for *TBP*. Representative results of a single experiment with n = 3 biological replicates, each one performed in technical duplicate, are shown as mean ± SD (***p-value<0.001). Three independent experiments were consistent. (**C**) Correlation between *Caspase-8* and *IL-6, IL-8, IL1β, MCP-1* and *VEGF* expression in human glioblastoma. Pearson correlation coefficients computed between gene expression profiles for *CASP-8, IL-6, IL-8, IL1β, MCP-1* and *VEGF*, in 174 glioblastoma RNA-Seq samples retrieved from the Cancer Genome Atlas. The correlation coefficient between expression profiles is proportional to the circle radii in the matrix, and additionally color-coded using the color scale reported to the right of the matrix. The plot was generated using the */corrplot/* R package (https://cran.r-project.org/web/packages/corrplot).

The following source data and figure supplements are available for figure 2:

**Source data 1.** Caspase-8 promotes the secretion of cytokines and growth factors.

**Source data 2.** Caspase-8 promotes mRNA expression of cytokines and growth factors.

**Source data 3.** Data collection for the analysis of the correlation between *Caspase-8* and *IL-6, IL-8, IL1β, MCP-1* and *VEGF* expression in human glioblastoma, *Figure 2C*.

**Source data 4.** Correlation between *Caspase-8* and *IL-6, IL-8, IL1β, MCP-1* and *VEGF* expression in human glioblastoma.

**Figure supplement 1.** Silencing of *Caspase-8* triggers the downregulation of *IL-6, IL-8* and *VEGF* mRNA expression.

**Figure supplement 2.** Ectopic expression of a dominant negative IKBα triggers the downregulation of *IL-6, IL-8, IL-1β, MCP-1* and *VEGF* mRNA expression.

(*Figure 4B*), suggesting that Caspase-8 exerts its function via NF-kB signalling, possibly through secreted cytokines. Indeed, as shown in *Figure 4C*, CM of control cells that endogenously express Caspase-8 was sufficient to restore resistance to TMZ in cells silenced for Caspase-8. Importantly, CM from IKBalphaSR-expressing cells, like that from Caspase-8-deficient cells, lost such a protective capability (*Figure 4D*). These results support the conclusion that Caspase-8 expression in GBM cells triggers resistance to TMZ via an autocrine loop and suggest that the level of Caspase-8 expression may correlate with prognosis. To test this hypothesis, we analyzed the survival of 77 high-glioma patients (of which 15 were censored), whose gene expression was measured using microarrays (*Phillips et al., 2006*). We classified the patients on their *Caspase-8* expression, dividing the *Caspase-8* expression distribution into three quantiles and considering the first quantile as low expression and the last quantile as high expression. Analysis of the survival curves shows that patients that have higher *Caspase-8* expression levels have a lower survival chance than those with lower expression (Chi-squared 10.5 on 1 degree of freedom, p-value 0.00117), supporting our hypothesis (*Figure 4E*). In addition, we investigated the survival rates of the same 77 patients classified into three distinct subtypes: mesenchymal (23 cases), proneural (30 cases) and proliferative (24 cases) tumors. Patients belonging to each subtype were stratified in terms of high and low *Caspase-8* expression, based on the distribution of *Caspase-8* probe intensities specific for each subtype. The significantly different *Caspase-8* expression levels in the proneural, mesenchymal and proliferative GBMs justified the use of different thresholds for each subtype. For patients classified as proneural, those having high *Caspase-8* expression show significantly lower survival probability (Chi-squared 7.2 on 1 degree of freedom, p-value 0.00732), while no difference was observed for the mesenchymal and proliferative subtypes (*Figure 4—figure supplement 2*).

Overall, we identify a novel function of Caspase-8 (*Figure 4F*), which supports a double agent role of Caspase-8 in cancer. The classical role of Caspase-8 in apoptosis may account for the correlation between loss of *Caspase-8* expression and unfavourable prognosis in medulloblastoma (*Pingoud-Meier et al., 2003*). Conversely, tumors that are strongly dependent on NFkB activity and cytokine production, such as GBM, may have a selective advantage in retaining *Caspase-8* expression. In these contexts, targeting of *Caspase-8* expression or of its tumorigenic functions represents a novel therapeutic approach.

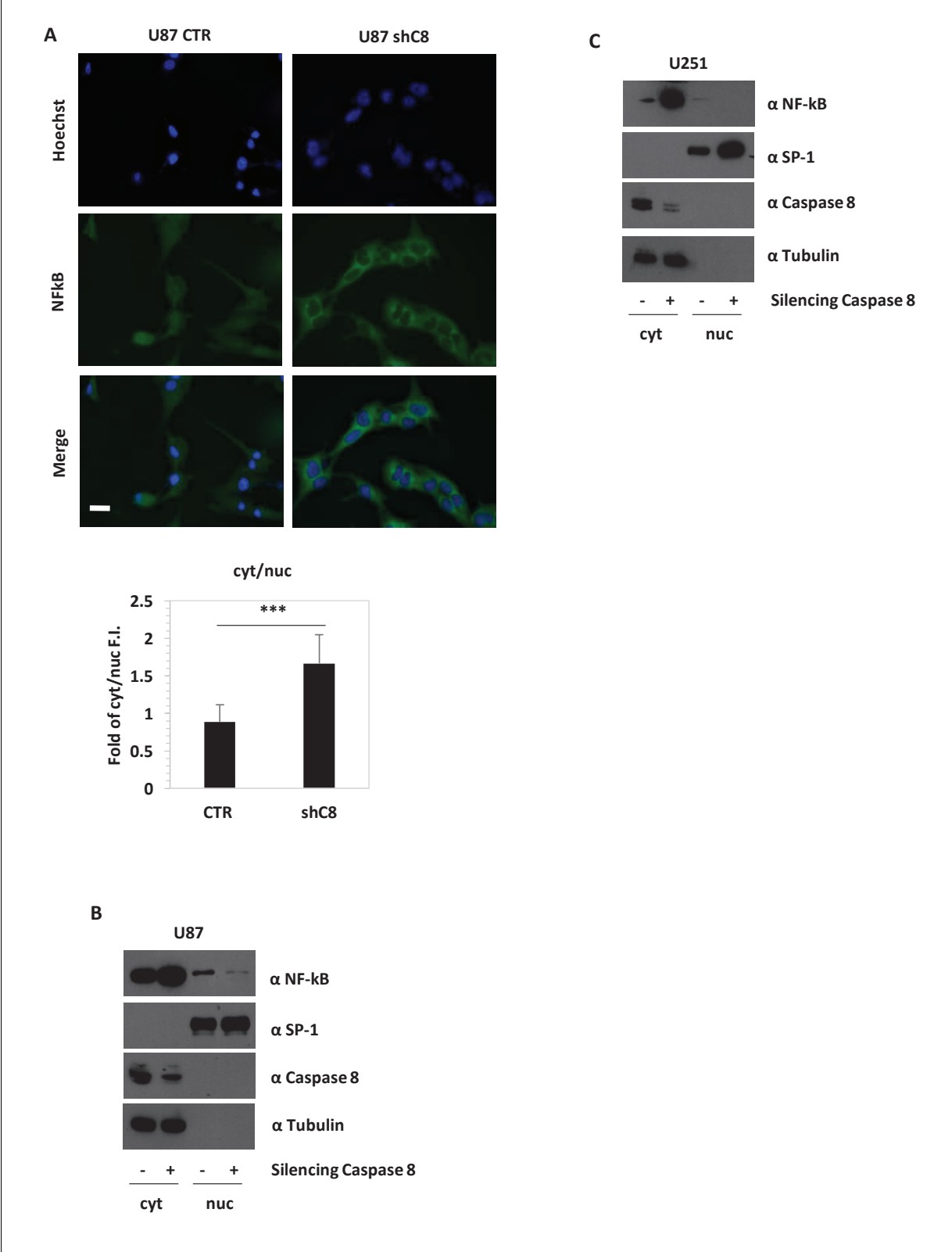

**Figure 3.** *Caspase-8* expression promotes NFkB translocation into the nucleus. (**A**) Immunostaining α-p65-NFkB in U87 Sh Caspase-8 and control cell lines. Bar scale: 25 μm. The bar chart represents the cytoplasmic (cyt)/nuclear (nuc) ratio of cell fluorescence intensity (FI) obtained using the Image J program. ***p-value<0.001. (**B, C**) Western blot analysis of fractionated cell lysates of U87 (**B**) and U251 (**C**) cell lines.

*Figure 3 continued on next page*

*Figure 3 continued*

The following source data and figure supplement are available for figure 3:

**Source data 1.** Caspase-8 promotes NFkB nuclear localization in U87 GBM cells.

**Source data 2.** Caspase-8 promotes *IL8* and *VEGFA* mRNA expression in tumors derived from mouse xenograft experiments.

**Figure supplement 1.** *Caspase-8* expression promotes NFkB activity in vivo.

## Materials and methods

### Cell cultures

*Caspase-8* was stably genetically silenced in U87 (originally obtained by ATCC) and U251 cell lines as previously described (*Fianco et al., 2016*). All cell lines were maintained in DMEM supplemented with 10% fetal bovine serum and were routinely tested negative for mycoplasma contamination. Following thawing, cells were used for no longer than one month.

Human umbilical endothelial cells (HUVEC; PromoCell GmbH, Heidelberg, Germany) were cultured as previously reported (*Gabellini et al., 2013*).

The sequences of *Caspase-8* used for interference are: shC8 5'-ATCACAGACTTTGGACAAA-3', shC8#2 5'-GCCTGGATGTTATTCCAG-3'; the sequence used as control, shcontrol (CTR), is: 5'-GGA TATCCCTCTAGATTA-3'.

The pMIGIKBalphaS32A/S36A (IKBalpha SR) construct was kindly provided by Y. Ciribilli and A. Inga (*Brockman et al., 1995*; *Bisio et al., 2014*).

### Antibodies and other reagents

Anti Caspase 8 (MBL 1:1000) RRID:AB_590760; anti Tubulin (Sigma-Aldrich 1:2000) RRID:AB_10013740; anti NF-kB (p65) (Santa Cruz 1:1000) RRID:AB_632037; anti SP1 (Santa Cruz 1:1000) RRID:AB_2171050; anti rat mAb CD31 (clone SZ31, Dianova GmbH 1:10) RRID:AB_2631039; anti-NF-kB p65 (clone E379, abcam, 1:1000) RRID:AB_776751; TMZ: temozolomide.

### In vitro and in vivo angiogenesis assays

All procedures involving animals and their care were authorized and certified by the decree n.26/2014 of the Italian Minister of Health following the relative guide lines.

For in vivo Matrigel assays, 60 μl 10× concentrated CM from the different cell lines, obtained using Centricon-3 concentrators (Merck Millipore, Billerica, MA) were mixed with 600 μl of Matrigel (BD Bioscience, San Jose, CA), supplemented with heparin (19.2 U; Schwarz Pharma SpA, Milan, Italy). This medium was injected subcutaneously into the flank of 8-week-old C57BL/6 mice (provided by the Animal Care Unit of the Regina Elena Cancer Institute, Rome, Italy). The negative and positive controls contained heparin alone or heparin plus VEGF (60 ng/mice; R&D Systems, Minneapolis, MN), respectively. After 5 days, the angiogenic response was evaluated by macroscopic analysis at autopsy, and by measurement of the hemoglobin (Hb) content in the pellet of matrigel as previously reported (*Gabellini et al., 2013*). The values were expressed as optical density (OD at 540 nm)/100 mg of matrigel. Each group consisted of ten animals. The experiments were repeated three times.

In vitro HUVEC cell proliferation was evaluated by a colorimetric assay at the end of treatment as described previously (*Gabellini et al., 2013*). Endothelial capillary tube-like network formation was assessed using Matrigel as described previously (*Gabellini et al., 2013*). Briefly, 24–well microtiter plates were coated with 300 μl/well unpolymerized matrigel (10 mg/ml) and allowed to polymerize at 37°C. Endothelial cells were plated ($5 \times 10^4$ cells/well) in 1 ml of serum free medium (negative control), complete medium (positive control), or conditioned medium (CM) obtained from sh control (CTR) or sh Caspase 8 (shC8) cells. After 8 hr, cell growth was observed through a reverted, phase-contrast photomicroscope and photographed. Experiments were repeated at least three times, and each sample was tested in triplicate. Angiogenic activity was quantified by measuring the cumulative length of the sprouts using digital imaging software (Image J) to analyze ten fields per experimental group and experiment.

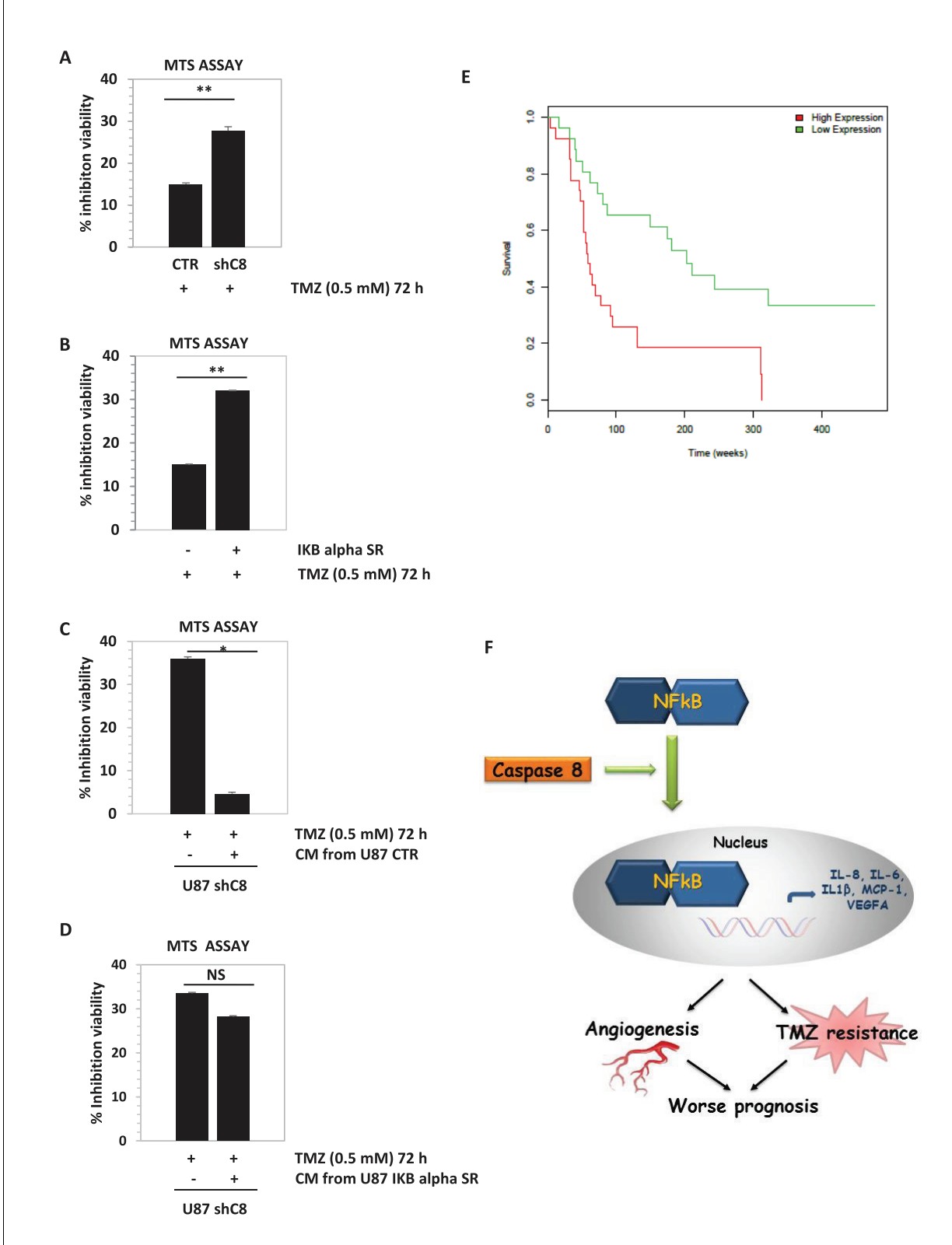

**Figure 4.** Downregulation of *Caspase-8* increases sensibility to Temozolomide (TMZ). (**A, B**) Viability assay represented in the histogram as mean ± SD. U87CTR control cell lines, U87shCaspase-8 and U87 IKBalphaSR, were incubated in the presence of Temozolomide (TMZ 0.5 mM for 72 hr) or not. The viability of TMZ-treated cells was assessed with the CellTiter 96 Aqueous One Solution Cell proliferation assay, and was represented as the percentage of inhibition of viability measured in cells without TMZ treatment. Data are represented in the histogram as mean ± SD. Error bars represent

*Figure 4 continued on next page*

*Figure 4 continued*

a SD between three (**A**) or two (**B**) independent experiments, each of them performed at least in technical triplicate. Student's *t* test was used for statistical analyses. **p<0.01 (**A**, **B**). (**C**, **D**) U87shCaspase-8 were incubated or not in the presence of Temozolomide (TMZ 0.5 mM for 72 hr) dissolved in conditioned media derived from U87 ShCtr (CTR), U87 shCaspase-8 (shC8) or U87IKBa SR. Data are represented in the histogram as mean ± SD. Error bars represent a SD between two (**C**) or three (**D**) independent experiments, each of them performed in technical triplicate. Student's *t* test was used for statistical analyses. *p-value<0.05 (**C**), NS=not significant. In all experiments shown in panel **A**–**D**, the volume of CM from different samples was normalized on the number on cells for each sample counted when the CM was collected. (**E**) Survival curves of high-grade glioma classified based on *Caspase-8* expression levels. Glioma patients were classified as low *Caspase-8* expression (green curve) and high Caspase-8 expression level (red curve), as described in the text. The Kaplan-Meier test supports a significant difference (p-value 0.00117) between the survival rates of the two groups, with patients having a low Caspase-8 expression showing a higher survival probability. (**F**) Proposed model depicting the link between Caspase-8 and cytokines in glioblastoma. Caspase-8 promotes NFkB nuclear localization and sustains the production of VEGF, IL-6, IL-8, IL-1$\beta$ and MCP-1. This pathway promotes neoangiogenesis and triggers resistance to Temozolomide.

The following source data and figure supplements are available for figure 4:

**Source data 1.** *Caspase-8* downregulation increases sensibility to Temozolomide (TMZ).

**Source data 2.** Survival curves of high-grade glioma classified on the basis of *Caspase-8* (*CASP8*) expression levels.

**Source data 3.** *Caspase-8* downregulation by two different shC8 constructs increases sensibility to Temozolomide (TMZ).

**Figure supplement 1.** *Caspase-8* downregulation by two independent interference sequences increases sensibility to Temozolomide (TMZ).

**Figure supplement 2.** Survival curves and *Caspase-8* expression levels in three GBM subtypes.

## Luminex assay

Concentrations of IL1-$\beta$, IL-6, IL-8, MCP-1 and VEGF-A were simultaneously determined in supernatants of shCtr (CTR) and ShC8 cells (n = 3 biological replicates per group and two technical duplicates for each sample) using a custom-made human magnetic Luminex assay kit (E-Bioscience). The assay procedure was performed according to the manufacturer's instructions and the plate was read on a Luminex-200 instrument (Luminex Corp., Austin, TX). Data were calculated by generating a calibration curve using the recombinant cytokines specified above, diluted in the cell culture medium used for culturing cells. Concentrations of each analyte were calculated using a standard 5P-logistic weighted curve generated for each target and expressed as picograms per milliliter (pg/ml). Due to out of range readings of undiluted samples, IL-8 concentrations were calculated on 1:10 diluted supernatants. Data were presented as the mean ± SD and the statistical analysis was performed using unpaired T-tests.

## Immunofluorescence

Cells were plated on coverslips and maintained at 37°C and 5% CO$_2$ for 24 hr before staining. Cells were treated or not with TNF$\alpha$ (20 ng/ml) 20 min before staining. Cells were washed with 1x phosphate buffer salin (PBS) three times. They were fixed in 4% paraformaldehyde for 15 min, permeabilized in 0.3% triton x100 for 15 min, blocked with 1% BSA for 1 hr at room temperature, and incubated with primary antibody overnight at 4°C. Secondary antibodies were applied for 1 hr at room temperature, stained with Hoechst for 5 min. The primary antibody used was anti NFkB p65 Santa Cruz (clone C20) 1:200. The secondary antibody was donkey anti rabbit 488 (Jackson Immune Research) 1:200. Images of immunostaining cells were obtained by microscopy using a Olympus BX53 microscope. Quantitative fluorescence data were exported from ImageJ generated histograms into Microsoft Excel software for further analysis and presentation. Cytoplasmic and nuclear staining intensities were compared to give the cytoplasmic/nuclear ratio.

## Reverse transcription and real time RT-PCR analyses

One microgram of total RNA isolated by TRIZOL reagent (Invitrogen, Carlsbad, CA, USA) was retro-transcribed with MLV-Reverse Transcriptase (Promega, Madison, WI, USA) according to standard procedures. Ten nanograms of cDNA were employed to quantify the transcripts by real time RT-PCR

using SYBR Select Master Mix (Applied Biosystem Foster City, CA, USA) and gene-specific primers, which are listed in supplemental information. Real-time PCR was performed using the 7900HT Fast Real-Time PCR System (Applied Biosystem). Relative quantity (RQ) was calculated normalizing for TBP and using a U87CTR samplesas calibrator. Mean values and standard deviations of RQ were generated from three biological replicates. Each experiment was performed for two technical replicates.

The following primer sequences were used:

**Primer sequences list**

| | |
|---|---|
| IL-6-FW | 5'- CAGGAGCCCAGCTATGAACT -3' |
| IL-6-REV | 5'-GAAGGCAGCAGGCAACAC- 3' |
| | |
| IL-8-FW | 5'-GGTGCAGTTTTGCCAAGGAG-3' |
| IL-8-RV | 5'-TGGGGTGGAAAGGTTTGGAG-3' |
| VEGF-FW | 5'-CCTTGCTGCTCTACCTCCAC-3' |
| VEGF-RV | 5'-CAACTTCGTGATGATTCTGC-3' |
| CCL2-MCP1-FW | 5'-CTTCATTCCCCAAGGGCTCG-3' |
| CCL2-MCP1-RV | 5'-GCTTCTTTGGGACACTTGCTG-3' |
| TNFA-FW | 5'-GGGACCTCTCTCTAATCAGC-3' |
| TNFA-RV | 5'-TCAGCTTGAGGGTTTGCTAC-3' |
| TBP-FW | 5'-TGCCCGAAACGCCGAATATAATC-3' |
| TBP RV | 5'-TGGTTCGTGGCTCTCTTATCCTC-3' |
| CASP8-FW | 5'- CAGCAGCCTTGAAGGAAGTC -3' |
| CASP8-RV | 5'-CGAGATTGTCATTACCCCACA-3' |

## Protein extracts and immunoblotting analysis

Cell extracts were prepared in IP buffer (50 mM Tris–HCl [pH 7.5], 250 mM NaCl, 1% NP-40, 5 mM EDTA, 5 mM EGTA, 1 mM phenylmethylsulfonyl fluoride, 25 mM NaF, 1 mM sodium orthovanadate, 10 µg/ml TPCK, 5 µg/ml TLCK, 1 µg/ml leupeptin, 10 µg/ml soybean trypsin inhibitor, 1 µg/ml aprotinin). For nuclei/cytoplasm cell fractionation, cells were in hypotonic buffer (10 mM HEPES [pH 7.5], 10 mM KCl, 0.1 mM EDTA, 0.1 mM EGTA, 1 mM DTT, and protease and phosphatase inhibitors at concentrations described below) and incubated for 15 min on ice. NP-40 (0.6% final concentration) was added, and nuclei were harvested by centrifugation at 12,000 g at 4°C for 30 s. The cytoplasmic fraction was recovered, and nuclear proteins were extracted from the pellet in nucleus buffer (20 mM HEPES [pH 7.5], 0.4M NaCl, 0.1 mM EDTA, 0.1 mM EGTA, 1 mM DTT, and protease and phosphatase inhibitors at concentrations described below) for 1 hr at 4°C on a rotating wheel. For immunoblotting, 50–100 µg of proteins were separated by sodium dodecyl sulfate (SDS) polyacrylamide gel electrophoresis (PAGE), blotted onto nitrocellulose membrane, and detected with specific antibodies.

## Cell viability analysis

U87 cell lines (U87 CTR and U87 Sh Caspase 8) were seeded in 96-well plates (1,000 cells/well) and treated for 72 hr with TMZ 0.5 mM or DMSO as control. Cell viability was analysed by CellTiter 96 Aqueous One Solution Cell proliferation assay (Promega) as previously described (*Stagni et al., 2015*).

## In vivo tumorigenic assay and angiogenesis analysis

For in vivo tumorigenicity, female CD-1 nude (nu/nu) mice, at 6–8 weeks old and 22–24 g in body weight, were purchased from Charles River Laboratories (Calco, Italy). $4 \times 10^6$ U87CTR and U87ShC8 cells were injected subcutaneously into the flank of these mice (24 for each group). Two different experiments (the first one with 8 and the second one with 16 animals for each group) were performed. The mice were observed daily, and their tumor volume (mm³) was calculated as length ×

width 2 × π/6. The animals were sacrificed 3 (CTR) and 6 (shC8) weeks after cell injection. The results were analysed by pooling together the two experiments to evaluate tumor growth at three weeks (CTR and shC8) and at six weeks (shC8). The three groups were compared using the Kruskal-Wallis test with Bonferroni correction. Immediately after sacrifice, the tumors were removed: half of each tumor was frozen in Trizol and stored at −80°C and the remaining half was fixed in 4% buffered formalin and paraffin embedded for immunohistochemical analysis.

Microvessel density and NFkB expression were evaluated on tumor xenograft paraffin-embedded sections by staining endothelial cells using a CD31 anti rat mAb and the rabbit mAb anti-NF-kB, respectively. Immunoreactions were revealed by ULTRATEK HRP (Scy Tek Laboratories, UT, USA) for CD31 and by Bond Polymer Refine Detection in an automated stainer (Leica Biosystem, Milan, Italy) for NFkB.

## Correlation study

A collection of 174 RNA-Seq data samples from patients diagnosed with glioblastoma multiforme was retrieved from the Cancer Genome Atlas (TCGA). The data were produced by the University of North Carolina Cancer Genomic Characterization Center (CGCC) using the Illumina HiSeq 2000 platform, and made available in TCGA as Level 3 (preprocessed) data. The TCGA data were retrieved from the Genomic Data Commons (GDC) using the following search query:

Disease Type IS Glioblastoma Multiforme AND Primary Site IS Brain AND Program Name IS TCGA AND Project Id IS TCGA-GBM AND Access IS open AND Data Category IS Gene expression AND Data Format IS TXT AND Data Type IS Gene expression quantification AND Experimental Strategy IS RNA-Seq AND Platform IS Illumina HiSeq

Among the files retrieved by this query, we employed the files reporting RSEM normalized gene expression. More details on how to retrieve these data are provided in the Supplementary Materials (*Supplementary file 1*). Data processing was carried out using the SeqWare Pipeline project's MapspliceRSEM workflow (version 0.7) (*O'Connor et al., 2010*). Gene-level expression data were estimated using RSEM (*Li and Dewey, 2011*) and normalized to set the upper quartile count at 1,000 for gene level. Correlation between gene expression profiles in the 174 samples was computed as the Pearson product-moment correlation coefficient, setting the p-value threshold at 0.01.

## Survival analysis on high-grade glioma patients

Microarray gene expression data, obtained using the Affymetrix U133A chip on a set of 77 high-grade gliomas (*Phillips et al., 2006*) and for which patient follow-up was available, were retrieved from the Gene Expression Omnibus (GEO series Id GSE4271) in the form of MAS5-normalized intensities. The distribution of *Caspase-8* expression was divided in three equal-size quantiles; patients whose *Caspase-8* expression was in the first quantile were classified as 'Low Expression', while those whose *Caspase-8* expression was in the third quantile were classified as 'High Expression'. Patient stratification depending on *Caspase-8* expression was computed for all patients together, and also independently for the three subtypes in which the patients were classified (proneural, mesenchymal or proliferative). Survival curves and the Kaplan-Meier estimator were computed and plotted using the R package *survival* (https://cran.r-project.org/web/packages/survival/index.html).

## Acknowledgements

We thank, D Stupack, Y Ciribilli, A Inga, for kindly providing reagents, V Stagni and F Moretti for critical reading of the manuscript and Gerry Melino for helpful discussion. This work has been supported by research grants from AIRC (IG1408, IG10590, n.19069), AICR (AICR 07–0461) and PRIN 2010_M4NEFY_005 to DB, and from AIRC (n. 18560) to DDB. MPM was supported by a fellowship provided by PNR-CNR Progetto Invecchiamento to AL.

## Additional information

### Funding

| Funder | Grant reference number | Author |
|---|---|---|
| Consiglio Nazionale delle Ricerche | PNR-CNR Progetto Invecchiamento | Maria Patrizia Mongiardi Andrea Levi |
| Associazione Italiana per la Ricerca sul Cancro | n. 18560 | Donatella Del Bufalo |
| Associazione Italiana per la Ricerca sul Cancro | IG1408 | Daniela Barilà |
| Ministero dell'Istruzione, dell'Università e della Ricerca | PRIN 2010_M4NEFY_005 | Daniela Barilà |
| Worldwide Cancer Research | AICR 07-0461 | Daniela Barilà |
| Associazione Italiana per la Ricerca sul Cancro | n.19069 | Daniela Barilà |
| Associazione Italiana per la Ricerca sul Cancro | IG10590 | Daniela Barilà |

The funders had no role in study design, data collection and interpretation, or the decision to submit the work for publication.

### Author contributions

GF, Designed and performed the cell biology and biochemical experiments, contributed to project conceptualization and co-wrote and edited the paper, Read and approved the final version of the paper; MPM, Contributed to Quantitative Real Time Experiments, and co-wrote and edited the paper, Read and approved the final version of the paper; AL, Contributed to project conceptualization, provided funding through competitively awarded grants to support MPM and co-wrote the paper, Read and approved the final version of the paper; TDL, Performed the neo-angiogenesis experiments in mice and their data analysis, Read and approved the final version of the paper; MD, Performed the in vivo tumorigenesis experiments (mouse xenograft experiments in mice) and their data analysis, Read and approved the final version of the paper; DT, Performed the in vitro angiogenesis experiments and their data analysis and co-wrote the paper, Read and approved the final version of the paper; DDB, Provided funding through competitively awarded grants, coordinated the angiogenesis experiments and co-wrote the paper, Read and approved the final version of the paper; IC, Set-up and performed the immunofluorescence experiments and their data analysis, Read and approved the final version of the paper; ADB, Set-up and performed the immunohistochemistry experiments on tumor samples and their data analysis, Read and approved the final version of the paper; MM, Supervisioned and performed the immunohistochemistry experiments on tumor samples and their data analysis, Read and approved the final version of the paper; AG, Set-up and performed the cytokine detection by Luminex Technology experiments and their data analysis, Read and approved the final version of the paper; DC, Supervisioned the cytokine detection by Luminex Technology experiments and their data analysis, Read and approved the final version of the paper; FF, Performed all the bioinformatic analysis and co-wrote the paper, Read and approved the final version of the paper; DB, Conceived the project, provided funding through competitively awarded grants, coordinated the project, supervisioned all the experiemnts and data analysis and wrote the paper, Read and approved the final version of the paper

### Author ORCIDs

Fabrizio Ferrè, http://orcid.org/0000-0003-2768-5305
Daniela Barilà, http://orcid.org/0000-0002-6192-1562

### Ethics

All procedures involving animals and their care were authorized and certified by the decree n.26/2014 of the Italian Minister of Health following the relative guide lines

## Additional files

### Supplementary files
• Supplementary file 1. The Cancer Genome Atlas data retrieval.

### Major datasets
The following previously published datasets were used:

| Author(s) | Year | Dataset title | Dataset URL | Database, license, and accessibility information |
|---|---|---|---|---|
| Phillips HS, Kharbanda S, Chen R, Forrest W, Soriano R, Wu TD, Misra A, Nigro J, Colman H, Soroceanu L, Williams PM, Modrusan Z, Feuerstein B, Aldape K | 2006 | Molecular subclasses of high-grade glioma: prognosis, disease progression, and neurogenesis | https://www.ncbi.nlm.nih.gov/geo/query/acc.cgi?acc=GSE4271 | Publicly available at the NCBI Gene Expression Omnibus (accession no: GSE4271) |

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
