## [Decision Letter]

Thank you for submitting your article "Caspase-8 contributes to angiogenesis and chemotherapy resistance in glioblastoma" for consideration by *eLife*. Your article has been reviewed by two peer reviewers, and the evaluation has been overseen by Jonathan Cooper as the Senior Editor and Reviewing Editor. The following individual involved in review of your submission has agreed to reveal his identity: Marcel Kool (Reviewer #3).

The reviewers have discussed the reviews with one another and the Reviewing Editor has drafted this decision to help you prepare a revised submission.

Summary:

The authors have studied the role of Caspase 8 in glioblastoma. Caspase 8 is a key player of extrinsic apoptosis and is often down-regulated in many cancers. However, Caspase 8 expression is often retained in glioblastoma. Following up on their previous findings that caspase 8 sustains neoplastic transformation in vitro, the authors now show that caspase 8 enhances expression and secretion of VEGF, IL6, IL8, IL1beta and MCP1 leading to increased neovascularization and increased resistance to temozolomide. They also show that high levels of caspase 8 in high grade gliomas are correlated with poor patient outcome.

The manuscript is well-written, figures are clear, and the findings presented are interesting, showing a new role for Caspase 8 in glioblastoma, different from its role in cancers where Caspase 8 is down-regulated and being mainly involved in apoptosis.

Required revisions:

There are a few important controls and other issues that must be addressed:

1) The manuscript presents two aspects of Casp-8 mediated functions (angiogenesis and TMZ resistance), both working through NFkB regulation of secreted factors. The main concern is lack of direct in vivo evidence. The addition of in vivo tumor angiogenesis (with staining for blood vessel content) would greatly improve the data (flank or orthotopic). It would also allow for more precise NFkB staining (with IHC) to show changes in localization.

2) It is important to show that the expression of Casp-8 is silenced by the shRNA in these cells either by protein or by mRNA analyses (preferably both).

3) To exclude off-target effects, the independent Casp-8 shRNA used in the supplementary figure must also be used for endothelial tube formation and TMZ experiments, and/or rescues done by expressing a Casp-8 cDNA that cannot be targeted by the shRNA.

4) Figure 1 and Figure 2 – The data presented show that shRNA targeting Casp-8 (shC8) decreases angiogenesis driven by GBM secreted factors in the assays used (Matrigel plug, endothelial tube formation), and causes down-regulation of associated secreted factors, which mostly correlate with gene expression data in human GBM samples. The impact of shC8 on GBM cell proliferation and survival during collection of the conditioned media (CM) should be quantified.

5) Figure 3 – Immunofluorescence staining for NFkB nuclear localization is not convincing and better data are needed.

6) Figure 4 – Analysis of GBM patient survival based on Casp-8 expression shows a significant impact on overall survival. Is the impact on survival due to elevated Casp-8 expression within the mesenchymal subgroup of GBM? The authors should comment on subgroup specific Casp-8 expression, and associated survival rates.

---

## [Author Response]

*Required revisions:*

*There are a few important controls and other issues that must be addressed:*

*1) The manuscript presents two aspects of Casp-8 mediated functions (angiogenesis and TMZ resistance), both working through NFkB regulation of secreted factors. The main concern is lack of direct in vivo evidence. The addition of in vivo tumor angiogenesis (with staining for blood vessel content) would greatly improve the data (flank or orthotopic). It would also allow for more precise NFkB staining (with IHC) to show changes in localization.*

To address this issue and uncover the in vivo relevance of Caspase-8 in GBM, mouse xenograft experiments were carried out as suggested by reviewers. We compared the tumorigenic potential of U87 cells stably interfered for Caspase-8 expression or not in nude mice xenograft experiments. Two experiments were performed: the first one with 8 animals for each group and the second one with 16 animals for each group. As reported in novel Figure 1, U87shC8 cells exhibited a drastically reduced capacity to form tumours as well as a significant reduction of tumor growth compared to U87 CTR cells.

To evaluate whether the decreased tumor growth observed in shC8 mice compared to CTR was associated to a lower vascular density, we evaluated neovascularization in CTR and shC8 tumors.

Tumor vessels were visualized using anti-CD31 mAb which permitted an assessment of the vascular density of the tumors. Figure 1 shows that there was a significant difference in the vascularization between CTR and shC8 tumors. Preliminary experiments on tumor samples, suggest that shC8 tumors show a different distribution of NFkB compared to CTR ones (Figure 3—figure supplement 1), consistently with results obtained on U87 cells; furthermore, we could show in the same samples a significant reduction of VEGF and IL-8 expression (Figure 3—figure supplement 1). Overall these data suggest that NFkB signaling may be compromised also in vivo. Future experiments will clarify this issue.

*2) It is important to show that the expression of Casp-8 is silenced by the shRNA in these cells either by protein or by mRNA analyses (preferably both).*

We apologize with the reviewer for not including in our previous version this information. Caspase-8 mRNA and protein expression in U87 cells interfered for Caspase-8 expression (with both shCasp8 sequences) or not, have been analyzed respectively by Quantitative Real Time RT-PCR and immunoblotting. These results clearly show that both our shCasp8 sequences are effective. We added this information in a novel figure named Figure 1—figure supplement 1.

*3) To exclude off-target effects, the independent Casp-8 shRNA used in the supplementary figure must also be used for endothelial tube formation and TMZ experiments, and/or rescues done by expressing a Casp-8 cDNA that cannot be targeted by the shRNA.*

As suggested by reviewers endothelial tube formation experiments as well as TMZ experiments including the independent Casp-8 shRNA (shC8#2) have been performed. The results obtained using shC8#2 were fully consistent with the ones obtained using the first shC8 sequence. The new TMZ experiment has been added as Figure 4—figure supplement 1.

All the experiments on endothelial cells have been moved in this revised version from Figure 2 (panels C, D, E in the original submission) to Figure 1—figure supplement 2 (panels A, B, C). We provide the images of the new experiment with shC8#2 to the reviewers (Figure 5).

Author response image 1.(**A**) Cell proliferation analysis of U87 sh CTR and U87 shC8 cell lines after starvation (16 or 24 hour) Cell number shave been counted in triplicate. Data are shown as mean ± SD. * Pvalue=0,04 (CTR/shC8 16 h starvation); *Pvalue=0,009 (CTR/shC8 24 h starvation (**B**) Viability assay represented in the histogram as mean ± s. U87CTR control cell lines, U87shCaspase 8 cell lines were incubated without serum (starvation) for 72 hours and viability was assessed with the CellTiter 96 Aqueous One Solution Cell Proliferation assay. Data are represented in the histogram as mean ± SD. Error bars represent a SD between technical triplicates. Student’s t test was used for statistical analysis. * P value=0,007**DOI:**
http://dx.doi.org/10.7554/eLife.22593.028

*4) Figure 1 and Figure 2 – The data presented show that shRNA targeting Casp-8 (shC8) decreases angiogenesis driven by GBM secreted factors in the assays used (Matrigel plug, endothelial tube formation), and causes down-regulation of associated secreted factors, which mostly correlate with gene expression data in human GBM samples. The impact of shC8 on GBM cell proliferation and survival during collection of the conditioned media (CM) should be quantified.*

We thank the reviewers for this observation that allowed us to better clarify an important issue of our experiments. As previously shown shC8 severely affects GBM cell proliferation in normal conditions (Fianco et. al, 2016, Exp Cell Research). According to this observation, as expected, we could indeed appreciate a decrease in cell proliferation and survival of shC8 cells compared to control ones during collection of the conditioned media (CM). We quantified this effect and we provide this information to the reviewers (Figure 6).

Author response image 2.HUVEC formed tube-like structures resembling a capillary plexus when exposed to conditioned media from U87 shcontrol cells (3156 ± 1474 mean of cumulative length of the sprouts ± SD), on the contrary when HUVEC are exposed to conditioned media collected from U87 shC8#2 cells, they are poorly organized (1105 ± 1960 mean of cumulative length of the sprouts ± SD) and most of the cells are rounded.**DOI:**
http://dx.doi.org/10.7554/eLife.22593.029

Being fully aware of this effect, all the experiment with CM performed in the original version of this manuscript as well as in the revised version were carried out normalizing the volume of CM of the different samples on the cell number of the different samples counted when collecting CM, in order to really compare the effect of different CMs (obtained from the same number of cells). We clarified this issue in the manuscript by adding a note in the Figure Legends.

*5) Figure 3 – Immunofluorescence staining for NFkB nuclear localization is not convincing and better data are needed.*

We performed several additional experiments and we always confirmed the same result. In addition, we quantified our results using Image J program to evaluate the Cytoplasmic/Nuclear ratio of cell fluorescence. A graph representing the Cytoplasmic/Nuclear ratio of cell fluorescence intensity using Image J program has been added in Figure 3.

*6) Figure 4 – Analysis of GBM patient survival based on Casp-8 expression shows a significant impact on overall survival. Is the impact on survival due to elevated Casp-8 expression within the mesenchymal subgroup of GBM? The authors should comment on subgroup specific Casp-8 expression, and associated survival rates.*

To evaluate this issue we investigated survival rates of the same 77 patients classified into three distinct subtypes: Mesenchymal (23 cases), Proneural (30 cases) and Proliferative (24 cases). Patients belonging to each subtype were stratified in high and low Caspase-8 expression, based on the distribution of Caspase-8 probe intensities specific for each subtype. We could observe a significantly difference of Caspase-8 expression levels between the Proneural and the Mesenchymal and Proliferative GBMs and therefore we used different thresholds for each subtype (Figure 4—figure supplement 2). For patients classified as Proneural (which are generally characterized by lower Caspase-8 expression), those having higher Caspase-8 expression show significantly lower survival chance (Chi-squared 7.2 on 1 degree of freedom, p-value 0.00732), while no difference was observed for the Mesenchymal and Proliferative subtypes. These data have been included in Figure 4—figure supplement 2. We can speculate that the effect of an increase of Caspase-8 expression level may be more relevant in the tumor subtype that is characterized by lower Caspase-8 expression levels. Future experiments will be required to 1) better clarify this issue extending the study to other GBM databases; 2) further clarify the molecular requirement for Caspase-8 function in GBM.